# Understanding the impact of fall armyworm (*Spodoptera frugiperda* J. E. Smith) leaf damage on maize yields

Chipo Chisonga[1,2], Gilson Chipabika[2], Philemon H. Sohati[2], Rhett D. Harrison[1] *

**1** CIFOR-ICRAF, Zambia Office, St Eugene Office Park, Lusaka, Zambia, **2** Department of Plant Science, School of Agriculture Sciences, University of Zambia, Lusaka, Zambia

\* r.harrison@cifor-icraf.org

## Abstract

Fall armyworm (*Spodoptera frugiperda* J. E. Smith), a serious pest of maize and other cereals, recently invaded the Old World potentially threatening the food security and incomes of millions of smallholder farmers. Being able to assess the impacts of a pest on yields is fundamental to developing Integrated Pest Management (IPM) approaches. Hence, working with an early maturing, medium maturing and late maturing variety, we inoculated maize plants with 2nd instar *S. frugiperda* larvae at V5, V8, V12, VT and R1 growth stages to investigate the effects of FAW induced damage on yield. Different plants were inoculated 0–3 times and larvae were removed after 1 or 2 weeks to generate a wide range of damage profiles. We scored plants for leaf damage at 3, 5 and 7 weeks after emergence (WAE) using the 9 point Davis scale. While at harvest we assessed ear damage (1–9 scale), and recorded plant height and grain yield per plant. We used Structural Equation Models to assess the direct effects of leaf damage on yield and indirect effects via plant height. For the early and medium maturing varieties leaf damage at 3 and 5 WAE, respectively, had significant negative linear effects on grain yield. In the late maturing variety, leaf damage at 7 WAE had an indirect effect on yield through a significant negative linear effect on plant height. However, despite the controlled screenhouse conditions, in all three varieties leaf damage explained less than 3% of the variation in yield at the plant level. Overall, these results indicate that *S. frugiperda* induced leaf damage has a slight but detectable impact on yield at a specific plant developmental stage, and our models will contribute to the development of decision-support tools for IPM. However, given the low average yields obtained by smallholders in sub-Saharan Africa and the relatively low levels of FAW induced leaf damage recorded in most areas, IPM strategies should focus on interventions aimed at improving plant vigour (e.g. through integrated soil fertility management) and the role of natural enemies, as these are likely to result in greater yield gains at lower cost than a focus on FAW control.

**Data Availability Statement:** All data and analytical scripts are available on Github in the following repository: https://github.com/rhettdharrison/ScreenhouseFAW.git

**Funding:** NORAD (#RAF-18/0013) for funding. The funders had no role in study design, data collection and analysis, decision to publish, or preparation of the manuscript.

**Competing interests:** The authors have declared that no competing interests exist.

# Introduction

Fall armyworm, *Spodoptera frugiperda* (J.E. Smith; Noctuidae, Lepidoptera), a voracious pest of maize native to the Americas [1], was first reported in West Africa in 2016 [2] and subsequently spread across Africa and Asia reaching Australia in 2018 [3]. The pest has been reported to feed on more than 350 host plants [4–7], but its main hosts are poaceous plants, including other staple crops such as wheat, sorghum and rice [1, 8]. *Spodoptera frugiperda* adult moths have a remarkable flight capability, which undoubtedly contributed to its rapid spread across the Old World. They can maintain self powered flight for over 24 hrs and cover over 100 km in a single flight [9, 10]. *Spodoptera frugiperda* is also highly fecund. Adult females can lay up to 1500 eggs during their adult lifespan of approximately three weeks and under optimal conditions ($\sim$25C) the moth takes about one month to complete its life-cycle [1, 8]. Hence, *S. frugiperda* can quickly colonise maize fields over an enormous area after the seedlings emerge, and then rapidly build up populations if not held in check by natural enemies.

When *S. frugiperda* was reported first in Africa in 2016, there was a paucity of knowledge on its management and, fearing worst case scenarios, governments released millions of dollars in emergency funding to procure and distribute chemical insecticides for its control. Many of these insecticides contained highly hazardous chemicals, such as monocrotophos and dichlorvos [11] and, furthermore, were often ineffective [12]. What is more, poor farmers in low income countries usually cannot afford personal protective clothing and do not understand how to use chemical pesticides safely, leading to high levels of exposure to toxic substances and accidental poisonings [13–16]. Widespread indiscriminate use of chemical pesticides also undermines the pest control services provided by natural enemies [17–19]. In its native range, *S. frugiperda* is attacked by a wide diversity of natural enemies, including over 250 species of insect parasitoids [20–24], as well as numerous predators, nematode parasites and entopathogenic fungi [23, 25, 26]. Already, a large diversity of natural enemies has already been recorded attacking *S. frugiperda* in the Old World [27–35]. However, there is abundant evidence that when chemical insecticides are applied to fields, natural enemies are negatively impacted and their efficacy as control agents declines drastically [19, 22]. Unfortunately, many African governments continue to subsidise chemical insecticides, at considerable cost to already stretched agricultural support budgets, despite increased understanding of the human and environmental risks.

It is a matter of urgency that governments reduce insecticide use and implement integrated pest management (IPM) strategies [36]. IPM emphasizes crop management practices and biological control (use of predators, parasitoids and entomopathogens), combined with monitoring and judicious use of safe insecticides. Application of insecticides is reserved as a last resort and preferably low toxicity, highly specific options, such as biological insecticides, are used to reduce impacts on natural enemies [37]. Use of insecticides–including both synthetic and biopesticides–under IPM is based on an understanding of economic injury levels (or action thresholds). That is, fields are surveyed for pest prevalence or plant damage and an estimate of the projected yield loss is made. The cost of an intervention is then computed to derive the cost/benefit ratio and determine whether it makes economic sense to apply an intervention, such as spraying a field. Hence IPM depends on having accurate models for predicting yield loss based on the infestation rate or plant damage observed. Unfortunately, understanding of *S. frugiperda* induced yield losses in maize are still poorly understood and currently used thresholds are little better than guesstimates [38, 39]. The purpose of this study was to contribute to developing models for predicting yield loss from information on *S. frugiperda* induced leaf damage assessments in maize.

## Materials and methods

### Location and time

The experiment was conducted at the University of Zambia field station (-15.39463$^o$ S, 280.33606$^o$ E; 1,263 m above sea level), Lusaka, Zambia from August 2021 to January 2022. The experiment was conducted in an irrigated screenhouse approximately 12 m x 25 m, which was fumigated with 'Boom Super 100 EC' (active ingredient Dichlorvos) at 100g l$^{-1}$ immediately after planting to prevent natural infestation of the experimental plants with *S. frugiperda* or control of the pest by parasitoids and other natural enemies. The soil in the screen house was analysed and found to be neutral (pH 6.9) with low organic matter (1.6%), N (0.13%), P (78.3 mg/kg) and K (0.34 cmol/kg), and moderate mineral contents. The soil was a loam with 42% sand, 41% silt and 17% clay.

### Maize varieties

Three maize varieties commonly used in Zambia were tested, including an early maturing (PAN 413), medium maturing (PAN 53) and late maturing (PAN 7M-83) variety.

### Experimental design

Plants were sown in rows (25 cm in-row: 0.75 inter-row spacing). Compound D fertilizer (20N: 10P: 20K: 5% S) was applied at planting at the rate of 200 kg ha$^{-1}$ and Urea (N: 46%) top dressing was applied at 4 weeks after plant emergence also at the rate of 200 kg ha$^{-1}$. The plants were drip watered. Thinning and gapping were conducted at 7 days after emergence to maintain plant density and the location of transplanted seedlings was recorded. There were nine rows per variety and each row included 1 replicate of each treatment. There were 27 treatments, including no-damage controls (243 plants). The position of each treatment within a row was randomised (i.e. each row corresponded to a block within a 'randomised complete block design') (Fig 1). Trials with each maize variety were run in parallel in different parts of the greenhouse. It was not possible to include the maize variety as a factor in a fully factorial design. Rather, we conducted three separate trials run in parallel. This was because each variety grows at a different rate and hence the varieties had to planted in separate plots to avoid complicating effects from competition for light and nutrients. And, there was insufficient space to replicate plots in the screenhouse.

### Experimental treatments

We inoculated plants with a single 2$^{nd}$ instar *S. frugiperda* larvae, which was placed in the maize funnel, at V5, V8, V12, VT and R1 maize growth stages. In half of the inoculations, the larvae were removed after 7 days to simulate larval death. Mature larvae or pupae were collected from the remaining inoculated plants after 14 days to prevent FAW from establishing a population inside the screenhouse. For the vegetative growth stages (V5, V8 and V12), individual plants were inoculated zero times (control), once, twice or three times. Thus, creating feeding damage at multiple times during plant growth in some treatments, and overall generating a wide range of exposure to *S. frugiperda* feeding among plants. A complete list of the treatments is given in S1 Table.

We elected to use 2$^{nd}$ instar larvae because (i) neonates disperse via ballooning and we wanted to ensure the innoculated plant received the treatment, and (ii) later instar larvae are more complicated to rear because FAW become cannibalistic from the 3$^{rd}$ instar (Hence, to ensure we had sufficient larvae to conduct the experiment we elected to use 2$^{nd}$ instar larvae).

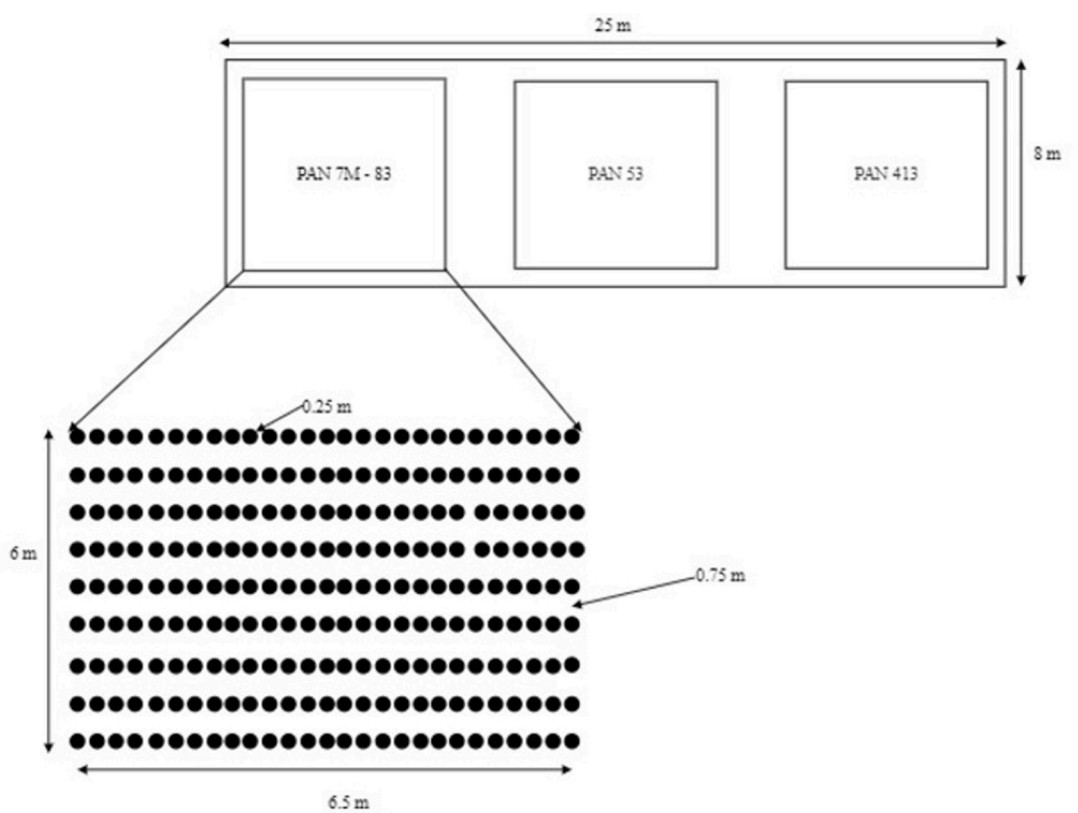

**Fig 1. Layout of the screenhouse experiment showing the position of the three trials (upper drawing), one on each of three varieties PAN 7M-83 (late maturing), PAN 53 (medium maturing), and PAN 413 (early maturing), and the detailed planting arrangement for one trial (lower drawing).** Maize was planted at 0.75 m inter-row and 0.25 m within-row spacing, which is the locally recommending plant spacing. Each trial consisted of nine rows and 27 plants along the row. Each plant in a row received a different treatment (S1 Table) and the allocation of treatments to plants was randomised within each row. Thus, each treatment was replicated once per row and the position of the treatment was randomised. Hence, each row comprised an experiment block, and in our models we used random intercepts for row and the plant position along each row to account for possible spatial autocorrelations in the response.

## Data collection

Leaf damage assessments of every plant were made at 3, 5 and 7 weeks after emergence, which corresponded to one week after inoculation for those plants that received a larva. The assessments were conducted by trained technicians, who were blind to the treatment of each particular plant thereby avoiding possible experimenter bias. On each occasion, on each plant the presence or absence of a *S. frugiperda* larva was recorded and the leaf damage was assessed using the Davis scale [40], which scores leaf damage 1–9, with 1 representing no damage and 9 being the most severe damage rating. Leaf damage was scored for the top three leaves and the whorl only to avoid counting damage from previous assessments. At harvest, plant height, number of cobs, cob length, cob damage (using the 9 point CIMMYT scale [41]), cob weight and the grain yield per plant were recorded. Before analysis, leaf and ear damage scores were rescale to 0–8, so that zero equalled no damage.

## Data analysis

Because FAW larvae sometimes moved to neighbouring plants, we could not use the assigned treatments for analysis. Instead, we calculated the exposure of each plant to *S. frugiperda* larval

feeding (1 or 2 weeks) at each growth stage. This also enabled us to calculate the lifetime exposure to larval feeding. From the assessments, we also obtained the leaf damage scores. Plots of cumulative exposure to *S. frugiperda* larvae versus cumulative leaf damage score, demonstrated that our treatments achieved a wide range of exposure and that this was closely correlated to the amount of leaf damage experienced by the plants (S1 Fig).

Plants that were recorded as transplants, wilting, attacked by Maize Streak Virus, or dead were removed from analysis. Correlation tests demonstrated that there was no association between exposure or leaf damage and these excluded categories. Ear damage was slight and had no significant effect on grain yield. Hence, to simplify the models we focused on the leaf damage results. Altogether out of the 243 plants established for each variety, 192 remained of the early maturing variety, 220 of the medium maturing variety and 226 of the late maturing variety.

All analyses were conducted in *R* v.4.2.1 [42]. For each variety separately, we employed Structural Equation Modeling using local estimation implemented in the packages *nlme* and *piecewiseSEM* v.2.1.2 [43] to elucidate the direct effects of *S. frugiperda* damage on grain yield and the indirect effects via plant height. The response was grain yield and we modelled the direct effects of plant height (cm), leaf damage (score 0–8) at 3, 5 and 7 weeks after emergence (score 0–8), as well as the indirect effects of leaf damage via plant height (Fig 2). We compared models with linear and polynomial functions for leaf damage score and considering leaf damage at 3, 5 and 7 weeks separately or as a combined effect by creating a composite variable [44]. Row number (block) and position along the row were entered as random effects. The variables were scaled to compare effect sizes. Models were simplified by removing the non-significant links and compared using Akaiki Information Criterion (AIC). We used the automatically reported model diagnostics including 'Tests of directed separation' and Fisher's C to assess model structure and overall model fit, respectively [43].

## Results

### Early maturing variety (PAN 413)

In the early maturing variety, there was no significant effect of leaf damage score at 3, 5 or 7 WAE on height and hence the SEM simplified to a linear mixed effects model. Leaf damage score at 5 WAE and 7 WAE also did not have any significant effect on yield, and were removed from the model. Thus, the best model, based on the AIC, retained just plant height and leaf damage score at 3 WAE (Table 1 and Fig 3A). Models with the combined effect of leaf damage at 3, 5, and 7 WAE did not perform any better than the model with leaf damage score at 3 WAE. The model with a linear function for leaf damage score performed better than the model with the polynomial function. Comparing models with and without leaf damage score at 3 WAE, the difference in marginal $r^2$ was 2.23%.

### Medium maturing variety (PAN 53)

In the medium maturing variety, as with the early maturing variety, there was no significant effect of leaf damage score at 3, 5 or 7 WAE on plant height and hence the SEM simplified to a linear mixed effects model. In the medium maturing variety, leaf damage score at 3 WAE and 7 WAE did not have any significant effect on yield. Hence, based on AIC, the best model retained just plant height and leaf damage score at 5 WAE (Table 2 and Fig 3B). Substituting leaf damage score at 5 WAE with combined leaf damage through a composite variable did not improve the model. The model with a linear function for leaf damage score performed better than one using a polynomial function. Comparing models with and without leaf damage score at 5 WAE, the difference in marginal $r^2$ was 2.21%.

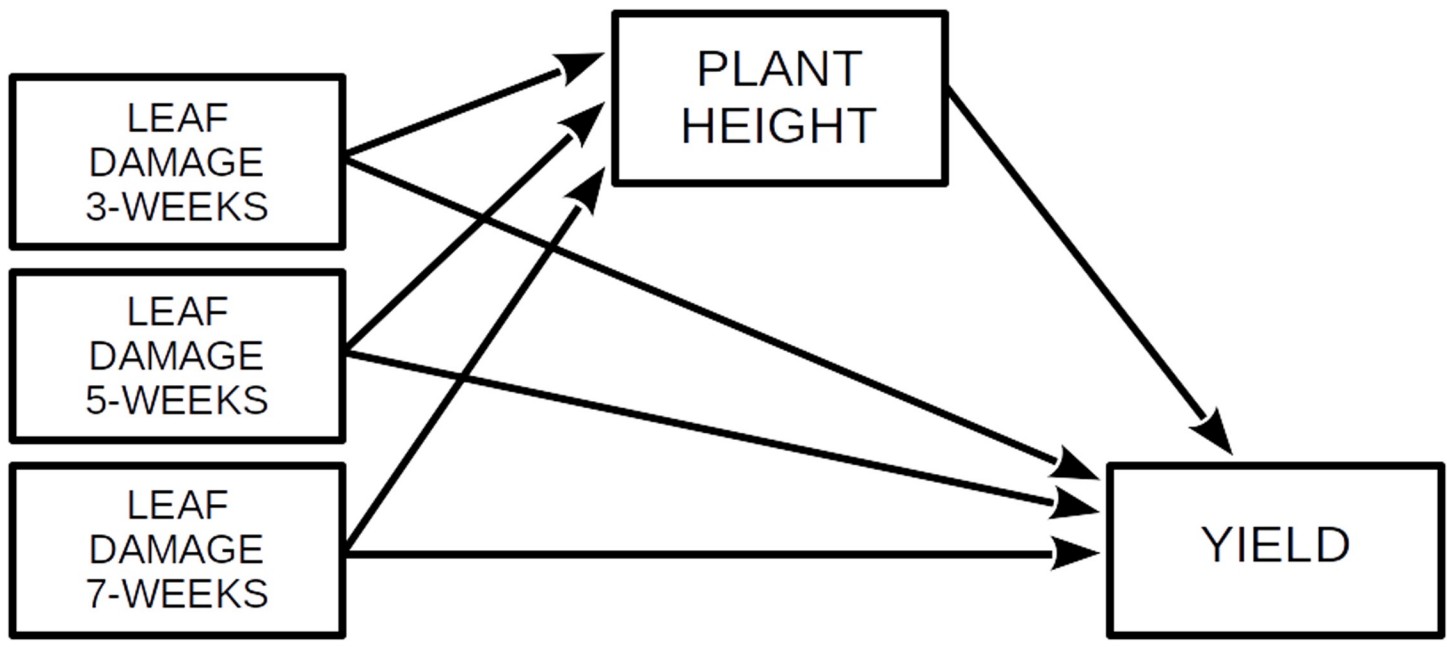

**Fig 2. Model design for structural equation model to investigate determinants of maize grain yield.** The full model was simplified by removing non-significant terms and comparing model AICs.

### Late maturing variety (PAN 7M-83)

In the late maturing variety, leaf damage score at 7 WAE had a significant negative effect on height but there was no direct effect of leaf damage score at 3, 5 or 7 WAE on grain yield. The best model for yield, based on the AIC, retained the direct effect of plant height and the indirect effect of leaf damage score at 7 WAE through plant height (Table 3 and Fig 3C and 3D). We also investigated substituting leaf damage score at 7 WAE with the combined leaf damage, but the model with leaf damage score at 7 WAE was the best performing. The model with a linear function for leaf damage score performed better than the one with a polynomial function. Comparing models for plant height with and without leaf damage score at 7 WAE, the difference in marginal $r^2$ was 1.62%.

### Discussion

Integrated Pest Management requires an understanding of the impacts of the pest on yields, so that informed decisions can be made by the farmer as to whether or not to apply insecticides. However, since *S. frugiperda* invaded the Old World, governments have spent millions of US dollars on providing subsidized pesticides to farmers without any proper understanding of the impacts of the pest on yields. Not only is this potentially wasting limited resources, but promoting the use of toxic chemicals to poor farmers, who rarely use protective clothing nor understand how to safely use poisons, carries substantial risks to human health and the environment.

**Table 1. Effect table for the best performing SEM model for grain yield in the early maturing variety (PAN 413).** Leaf damage was assessed on the nine point Davis scale [40] (score 0–8;). There were 162 degrees of freedom. Marginal r2 for leaf damage = 0.02.

| Response—Fixed effect | Estimate | Std. error | crit-value | p-value | Std. estimate |
|---|---|---|---|---|---|
| Grain weight–Plant height | 1.31 | 0.141 | 9.39 | 0.0000 | 0.5867 |
| Grain weight–Leaf damage 3 WAE | -8.67 | 3.351 | -2.58 | 0.0105 | -0.1467 |

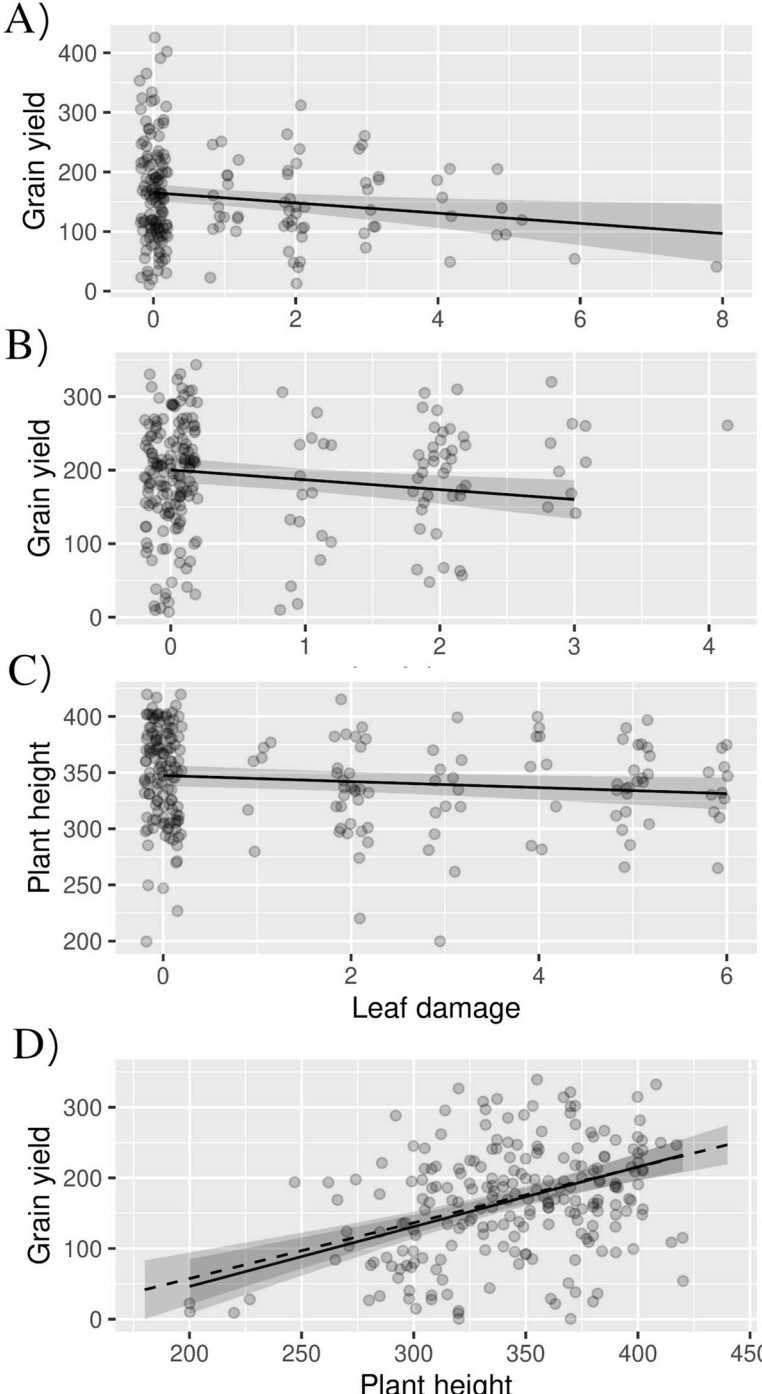

**Fig 3. Impact of fall armyworm leaf damage on yield.** Points indicate the observations, while the trend lines indicate the model predictions and the grey area their 95% confidence envelope. Only the significant interactions are illustrated. A) Grain yield against leaf damage at 3 WAE in the Early maturing variety. B) Grain yield agains leaf damage at 5 WAE in the Medium maturing variety. C) Plant height at harvest again leaf damage at 7 WAE in the Late maturing variety. D) Grain yield against plant height at harvest in the Late maturing variety, showing the effect of plant height alone (solid line) and the effect of plant height after controlling for the effect of leaf damage on plant height (dashed line).

**Table 2. Effect table for the best performing model for grain yield in the medium maturing variety (PAN 53).** Leaf damage was assessed on the nine point Davis scale [40] (score 0–8). There were 190 degrees of freedom. Marginal r2 for leaf damage = <0.01.

| Response—Fixed effect | Estimate | Std. error | Crit-value | p-value | Std. estimate |
|---|---|---|---|---|---|
| Grain weight–Plant height | 0.797 | 0.1156 | 6.894 | 0.0000 | 0.4351 |
| Grain weight–Leaf damage at 5 WAE | -12.705 | 4.9766 | -2.553 | 0.0115 | -0.1533 |

Chemical pesticides also often have greater impact on non-target organisms, including natural enemies, than on the pest, leading to perverse outcomes for pest control [17, 19, 45].

Current recommendations to smallholder farmers in Africa suggest that they should apply pesticides if *S. frugiperda* infestations exceed 20% of plants during the early season or 40% during the late season [38]. However, infestation is a poor indicator of yield loss, because natural mortality, driven by natural enemies and other factors such as rainfall [46], is highly variable. Hence, infestation level imparts very little useful information [47]. Assessments based on leaf damage hold more promise, although maize can also recover from high levels of damage [48, 49].

To improve understanding of the relationship between leaf damage and yield loss, we performed a screenhouse experiment on three commonly used varieties in Zambia In all three varieties we found a significant impact of leaf damage score on yield at a specific growth phase. Moreover, the sensitive growth phase was correlated with the duration of maturation; thus it was 3 WAE in the early maturing variety, 5 WAE in the medium maturing variety and 7 WAE in the late maturing variety. Together these results indicate that the leaf damage score assessed through scouting a field can be used to estimate yield loss, but only if the assessment is made during the period when the maize is sensitive to leaf damage, which varies with the maturation rate.

It is also worth stressing that, despite the screen-house conditions and the fact that we removed transplants, dead or wilting plants and those infected with Maize Streak Virus from the sample, leaf damage score explained only a very small proportion of the variance (<3%) in grain yield. Nevertheless, this is comparable to results from other controlled experiments on the impacts of FAW on yield [50, 51]. Moreover, in smallholders' fields there are plot scale factors such as differences in soil fertility, soil water holding capacity and competition from weeds, that have even greater effects on grain yield. Hence, from a farm systems perspective, improving soil management and crop husbandry among smallholders is likely to lead to much greater yield benefits at lower cost than a focus on FAW control [41]. In addition, in advising farmers on appropriate maize varieties to plant, the focus should be on those varieties that demonstrate good performance under smallholder management, such as drought tolerance and adequate growth in poor soils. Typically pest resistance carries a yield penalty [50, 51] and hence the use of *S. frugiperda* resistant or tolerant varieties may only be justified if the anticipated pest damage is very high, or if these attributes can be linked with others that enhance performance under smallholder management.

Assessing thresholds for action, such as applying a pesticide, under IPM is complex because of the need to integrate information on the current impact of the pest and the expected future impact of the pest as the crop matures. Our experiment shows that leaf damage at a specific developmental phase has a slight but significant negative impact on yield. Leaf damage at other

**Table 3. Effect table for the best performing model for grain yield in the late maturing variety (PAN 7M-83).** Leaf damage was assessed on the nine point Davis scale [40] (score 0–8). There were 198 degrees of freedom. Marginal r2 for leaf damage = <0.02.

| Response—Fixed effect | Value | Std. error | critical-value | p-value | Std. estimate |
|---|---|---|---|---|---|
| Grain weight–Plant height | 0.837 | 0.1317 | 6.352 | 0.0000 | 0.4074 |
| Plant height–Leaf damage at 7 WAE | -2.689 | 1.2829 | -2.0958 | 0.0374 | -0.1282 |

developmental phases does not impact yield and therefore can be ignored. However, FAW can also impact yields through feeding on the maize ears and can in addition spoil the grain through aflatoxin infection. The degree to which yield loss from ear damage can be predicted from leaf damage measured earlier in the crop cycle is still currently limited. In an on-farm experiment measuring FAW impact across 12 landscapes in Malawi and Zambia [47], FAW incidence and leaf damage declined from 3 WAE to 6 WAE and both leaf damage and ear damage were limited. However, where FAW populations are very high, they could potentially escape natural mortality factors if natural enemy populations are unable to track pest populations. Under such conditions, the FAW population could increase rapidly and cause substantial damage to maize ears. These thresholds are still not well understood [39], but a focus on the proportion of plants with severe leaf damage or infestation with late stage larvae (e.g. 4-6th instars) may be informative, because these metrics provide an index of the number of FAW larvae surviving to the late stages [47].

## Conclusions

We conducted a screen-house experiment to elucidate the impact of leaf damage on grain yield in maize, using commonly used varieties in Zambia. We found that leaf damage scores, obtained through scouting, could be used to predict yield loss. However, plants were only sensitive to leaf damage for a narrow window of time which correlated with the maturation rate of the variety. We also found that, despite the controlled conditions, leaf damage score explained only a small proportion of the variance in grain yield. Hence, in advising smallholder farmers, greater returns are likely to come from improved soil management and crop husbandry, than from optimizing *S. frugiperda* control. Nevertheless, our results will contribute to developing IPM tools for assessing potential yield loss and thus provide farmers with the information they require to make decisions on pest control interventions.

## Supporting information

**S1 Fig. Cumulative leaf damage against the cumulative number of weeks exposure to fall armyworm larval feeding.** According to different treatments, plants were innocualted with 2nd instar fall armyworm larvae at 2, 4 and 6 weeks after seedling emergence, and some larvae were removed after 1 week while others were allowed to feed for 2 weeks, which was sufficient to reach the 6th instar or pupae stages. Larvae sometimes migrated to neighbouring plants, hence the data shown are the observed weeks of exposure to larval feeding. Leaf damage was assessed at 3, 5 and 7 weeks after seedling emergence using the Davis 9 point scale, which was rescale from 0–8. A) Early maturing variety; B) Medium maturing variety; C) Late maturing variety.
(DOCX)

**S1 Table. Summary of treatments.** The V5, V8, V12, VT and R1 indicate the maize stage. L = Larval inoculation. RM indicates larvae were removed after 7 days.
(DOCX)

## Acknowledgments

The authors would like to thank the University of Zambia, Department of Agriculture for use of the screenhouse.

## Author Contributions

**Conceptualization:** Rhett D. Harrison.

**Funding acquisition:** Rhett D. Harrison.

**Investigation:** Chipo Chisonga, Philemon H. Sohati.

**Methodology:** Rhett D. Harrison.

**Supervision:** Gilson Chipabika, Rhett D. Harrison.

**Writing – original draft:** Chipo Chisonga.

**Writing – review & editing:** Rhett D. Harrison.

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
