## [Decision Letter · Decision Letter 0]

27 Feb 2023

PONE-D-22-30949Understanding the impact of fall armyworm (Spodoptera frugiperda J. E. Smith) leaf damage

on maize yieldsPLOS ONE

Dear Dr. Harrison,

Thank you for submitting your manuscript to PLOS ONE. After careful consideration, we feel that it has merit but does not fully meet PLOS ONE’s publication criteria as it currently stands. Therefore, we invite you to submit a revised version of the manuscript that addresses the points raised during the review process. Please make sure to address carefully all points raised by the two reviewers.

We look forward to receiving your revised manuscript.

Kind regards,

Nicolas Desneux

Academic Editor

PLOS ONE

Journal Requirements:

2. "PLOS requires an ORCID iD for the corresponding author in Editorial Manager on papers submitted after December 6th, 2016. Please ensure that you have an ORCID iD and that it is validated in Editorial Manager. To do this, go to ‘Update my Information’ (in the upper left-hand corner of the main menu), and click on the Fetch/Validate link next to the ORCID field. This will take you to the ORCID site and allow you to create a new iD or authenticate a pre-existing iD in Editorial Manager. Please see the following video for instructions on linking an ORCID iD to your Editorial Manager account: " ext-link-type="uri" xlink:type="simple">https://www.youtube.com/watch?v=_xcclfuvtxQ"

3. We note that you have stated that you will provide repository information for your data at acceptance. Should your manuscript be accepted for publication, we will hold it until you provide the relevant accession numbers or DOIs necessary to access your data. If you wish to make changes to your Data Availability statement, please describe these changes in your cover letter and we will update your Data Availability statement to reflect the information you provide

Reviewers' comments:

Reviewer's Responses to Questions

**Comments to the Author**

1. Is the manuscript technically sound, and do the data support the conclusions?

Reviewer #1: Yes

Reviewer #2: Yes

2. Has the statistical analysis been performed appropriately and rigorously? 

Reviewer #1: Yes

Reviewer #2: Yes

3. Have the authors made all data underlying the findings in their manuscript fully available?

Reviewer #1: Yes

Reviewer #2: Yes

4. Is the manuscript presented in an intelligible fashion and written in standard English?

Reviewer #1: Yes

Reviewer #2: No

5. Review Comments to the Author

Reviewer #1: The paper is well written and well executed. I recommend the following changes before its definitive acceptance:

Abstract

At lines 32-33, OK looking at the leaf damage at 3, 5 and 7 weeks after plant emergence (WAE) but corresponding to which maize plants growth stages? V5, V8, V12, VT and R1 ? I suggest to precise that in the abstract.

Mat. Met

It would be nice and helpful to get a scheme of the experimental design described.

Please for the experimental treatment, at line 131, explain us why 2nd instar of FAW and not other larval stage such as neonates?

The part is not clear from line 135 to 139: “For the vegetative growth stages (V5, V8 and V12), plants were inoculated zero times (control), once, twice or three times in different treatments. Thus, creating feeding damage at multiple times during plant growth in some treatments, and overall generating a wide range of exposure to S. frugiperda feeding among plants. » can you rephrase or change it to make it clearer?

Also at line 149, the authors say that “the chlorophyll content were recorded” but by which mean? technique? This is not explained. Moreover, there is no data shown on those results in the result section? Then better to avoid to precise that if it is not shown.

Results

It is missing the comparisons or the study on the influence of maize variety. The authors presented one by one each variety, but another sub-section would be nice to add to check statistically the influence of maturity characteristics on the data measured. The authors discussed on it later but we expect to have some statistics on the influence of the variety used.

Discussion

At line 137, the following part “We inoculated plants with plants at several developmental stages (V5, V8, V12, VT and R1 maize growth stages) and plants were inoculated from zero to three times. In addition, on half the inoculated plants the larvae were removed after 1 week to simulate larval death. This led to a wide range of exposure to larval feeding and cumulative leaf damage (Supplementary Materials, Figure S1).” Is a repetition to what it has been described in Mat. Met. section then consider deleting it.

Reviewer #2: Chisonga et al. investigated the impact of fall armyworm, Spodoptera frugiperda (J. E. Smith) leaf damage on maize yields by inoculating 2nd instar S. frugiperda larvae at V5, V8, V12, VT and R1 growth stages of early maturing (PAN 413), medium maturing (PAN 53) and late maturing (PAN 7M-83) varieties. Overall, the results showed that S. frugiperda induced slight leaf damage but detectable impact on yield at a specific plant developmental stage. This model will contribute to the development of decision-support tools for IPM. Overall, the topic is interesting and the data is well enough to support the main conclusion. However, the manuscript needs careful proofreading and revision. Grammar mistakes and poor writing are undermining the significance of this study. Besides, there are some major points which need clarifications before possible publication in PONE.

Introduction section

In line 59, I suggest providing more examples of studies describing host range and invasiveness of Spodoptera frugiperda, for example recent studies by Sokame et al. (2020), Wu et al. (2021) and Wang et al. (2022) can be cited.

* Sokame BM, Subramanian S, Kilalo DC et al. (2020) Larval dispersal of the invasive fall armyworm, Spodoptera frugiperda, the exotic stemborer Chilo partellus, and indigenous maize stemborers in Africa. Entomol Exp Appl 168:322–331.

* Wu PX, Ren QL, Wang W, Ma Z, Zhang RZ (2021a) A bet-hedging strategy rather than just a classic fast life-history strategy exhibited by invasive fall armyworm. Entomol Gen 4:337–344.

* Wang, P.; He, P.C.; Hu, L.; Chi, X.L.; Keller, M.A.; Chu, D. (2022) Host selection and adaptation of the invasive pest Spodoptera frugiperda to indica and japonica rice cultivars. Entomol. Gen. 42, 403–411.

In introduction section, a reference is missing to support the statement made is line 77, please revise as …provided by natural enemies (Croft 1990; Desneux et al. 2007).

- Croft, B. A. (1990). Arthropod biological control agents and pesticides. John Wiley and Sons Inc..

- Desneux, N., et al., 2007. The sublethal effects of pesticides on beneficial arthropods. Annu. Rev. Entomol. 52, 81–106.

- L118: The authors should provide a schematic for the methodology section, which will help readers to easily understand the overall sketch of this study.

- “We inoculated plants with a single 2nd instar S. frugiperda larvae, which was placed in the maize funnel, at V5, V8, V12, VT and R1 maize growth stages” Why second instar? Please discuss it.

- “In addition, the height of the plant and the chlorophyll content were recorded.” How? Please add further details about this statement.

- The data analysis section too much wordy and complicated. The authors should rewrite more clearly and concisely.

- In Discussion Section: “Not only is this potentially wasting limited resources, but promoting the use of toxic chemicals to poor farmers, who rarely use protective clothing nor understand how to safely use poisons, carries substantial risks to human health and the environment.” This sentence is very important. The authors should also add that the overuse/misapplication of chemicals insecticides have side effects on non-target species (Desneux et al. 2007; Palma-Onetto et al. 2021). Please cite these key articles.

- Desneux, N., et al., 2007. The sublethal effects of pesticides on beneficial arthropods. Annu. Rev. Entomol. 52, 81–106.

- Palma-Onetto, V., Oliva, D., González-Teuber, M. (2021). Lethal and oxidative stress side effects of organic and synthetic pesticides on the insect scale predator Rhyzobius lophanthae. Entomologia Generalis, 41 (4), 345-355.

- From sentences “To improve understanding of the relationship between leaf damage and yield loss, we performed a screenhouse experiment on three commonly used varieties in Zambia, including an early, medium and late maturing variety.” To “Typically pest resistance carries a yield penalty [44,45] and hence the use of S. frugiperda resistant or tolerant varieties may only be justified if the anticipated pest damage is very high, or if these attributes can be linked with others that enhance performance under small holder management” Both these paragraphs are redundant. The results are already presented in results sections. In discussion, the authors should compare and justify their main findings to recently published work.

- The current version of discussion section is not okay. The authors should rewrite whole discussion section. Remove the redundant sentences, compare and justify your results with recently published related work from reputable journals.

- The authors should merge the A, B, C, and D separated figures as Figure 2.

References:

The ref 1 is outdated and should be deleted because the new reference by Kenis et al. (2023) is highly comprehensive and detailed.

L315, citation for reference 2 should be corrected as :

Kenis M, Benelli G, Biondi A, Calatayud PA, Day R, et al. (2023) Invasiveness, biology, ecology, and management of the fall armyworm, Spodoptera frugiperda. Entomologia Generalis doi.org/10.1127/entomologia/2022/1659

6. PLOS authors have the option to publish the peer review history of their article (what does this mean?). If published, this will include your full peer review and any attached files.

Reviewer #1: No

Reviewer #2: No

---

## [Author Response · Author response to Decision Letter 0]

24 May 2023

PONE-D-22-30949

Understanding the impact of fall armyworm (Spodoptera frugiperda J. E. Smith) leaf damage

on maize yields

PLOS ONE

Dear Dr. Harrison,

Thank you for submitting your manuscript to PLOS ONE. After careful consideration, we feel that it has merit but does not fully meet PLOS ONE’s publication criteria as it currently stands. Therefore, we invite you to submit a revised version of the manuscript that addresses the points raised during the review process.

Please make sure to address carefully all points raised by the two reviewers.

If applicable, we recommend that you deposit your laboratory protocols in protocols.io to enhance the reproducibility of your results. Protocols.io assigns your protocol its own identifier (DOI) so that it can be cited independently in the future. For instructions see: https://journals.plos.org/plosone/s/submission-guidelines#loc-laboratory-protocols. Additionally, PLOS ONE offers an option for publishing peer-reviewed Lab Protocol articles, which describe protocols hosted on protocols.io. Read more information on sharing protocols at https://plos.org/protocols?utm_medium=editorial-emailutm_source=authorlettersutm_campaign=protocols.

We look forward to receiving your revised manuscript.

Kind regards,

Nicolas Desneux

Academic Editor

PLOS ONE

Journal Requirements:

We have updated the figures to meet the style guidelines.

2. "PLOS requires an ORCID iD for the corresponding author in Editorial Manager on papers submitted after December 6th, 2016. Please ensure that you have an ORCID iD and that it is validated in Editorial Manager. To do this, go to ‘Update my Information’ (in the upper left-hand corner of the main menu), and click on the Fetch/Validate link next to the ORCID field. This will take you to the ORCID site and allow you to create a new iD or authenticate a pre-existing iD in Editorial Manager. Please see the following video for instructions on linking an ORCID iD to your Editorial Manager account: https://www.youtube.com/watch?v=_xcclfuvtxQ"

We have linked the Orchid acc to the PONE account

3. We note that you have stated that you will provide repository information for your data at acceptance. Should your manuscript be accepted for publication, we will hold it until you provide the relevant accession numbers or DOIs necessary to access your data. If you wish to make changes to your Data Availability statement, please describe these changes in your cover letter and we will update your Data Availability statement to reflect the information you provide

We have archived the data and the analytic scripts and provided the information in the Cover Letter

We note this requirement.

Reviewers' comments:

Reviewer's Responses to Questions

Comments to the Author

1. Is the manuscript technically sound, and do the data support the conclusions?

Reviewer #1: Yes

Reviewer #2: Yes

2. Has the statistical analysis been performed appropriately and rigorously?

Reviewer #1: Yes

Reviewer #2: Yes

3. Have the authors made all data underlying the findings in their manuscript fully available?

Reviewer #1: Yes

Reviewer #2: Yes

4. Is the manuscript presented in an intelligible fashion and written in standard English?

Reviewer #1: Yes

Reviewer #2: No

5. Review Comments to the Author

Reviewer #1: The paper is well written and well executed. I recommend the following changes before its definitive acceptance:

Abstract

At lines 32-33, OK looking at the leaf damage at 3, 5 and 7 weeks after plant emergence (WAE) but corresponding to which maize plants growth stages? V5, V8, V12, VT and R1 ? I suggest to precise that in the abstract.

 We have clarified the text at this point.

Mat. Met

It would be nice and helpful to get a scheme of the experimental design described.

We now provide a scheme of the experimental design (Figure 1)

Please for the experimental treatment, at line 131, explain us why 2nd instar of FAW and not other larval stage such as neonates?

 We now provide a explanation, as follows: “We elected to use 2nd instar larvae because (i) neonates disperse via ballooning and we wanted to be sure the innoculated plant received the treatment, and (ii) later instar larvae are more complicated to rear because FAW become cannibalistic from the 3rd instar.”

The part is not clear from line 135 to 139: “For the vegetative growth stages (V5, V8 and V12), plants were inoculated zero times (control), once, twice or three times in different treatments. Thus, creating feeding damage at multiple times during plant growth in some treatments, and overall generating a wide range of exposure to S. frugiperda feeding among plants. » can you rephrase or change it to make it clearer?

We have inserted the word ‘individual’ to clarify the meaning: “For the vegetative growth stages (V5, V8 and V12), individual plants were inoculated zero times (control), once, twice or three times”

Also at line 149, the authors say that “the chlorophyll content were recorded” but by which mean? technique? This is not explained. Moreover, there is no data shown on those results in the result section? Then better to avoid to precise that if it is not shown.

We elected not to use these data, so we have removed the offending sentence. 

Results

It is missing the comparisons or the study on the influence of maize variety. The authors presented one by one each variety, but another sub-section would be nice to add to check statistically the influence of maturity characteristics on the data measured. The authors discussed on it later but we expect to have some statistics on the influence of the variety used.

We cannot explicitly compare among varieties, because variety was not included as factor in the experimental design. As explained in the methods section, it was necessary to grow each variety separately because of their different growth rates (which if allowed would have introduced other complications such as light competition). And, there was insufficient room in the screenhouse to replicate at the variety level. We have included more text in the methods section to explain this limitation. “It was not possible to include the maize variety as a factor in a fully factorial design. Rather, we conducted three separate trials run in parallel. This was because each variety grows at a different rate and hence the varieties had to planted in separate plots to avoid complicating effects from competition for light and nutrients. And, there was insufficient space to replicate plots in the screenhouse.”

Discussion

At line 137, the following part “We inoculated plants with plants at several developmental stages (V5, V8, V12, VT and R1 maize growth stages) and plants were inoculated from zero to three times. In addition, on half the inoculated plants the larvae were removed after 1 week to simulate larval death. This led to a wide range of exposure to larval feeding and cumulative leaf damage (Supplementary Materials, Figure S1).” Is a repetition to what it has been described in Mat. Met. section then consider deleting it.

We removed the text in parentheses to shorten it. However, we believe that briefly repeating a description of the experiment helps to explain our findings.

Reviewer #2: Chisonga et al. investigated the impact of fall armyworm, Spodoptera frugiperda (J. E. Smith) leaf damage on maize yields by inoculating 2nd instar S. frugiperda larvae at V5, V8, V12, VT and R1 growth stages of early maturing (PAN 413), medium maturing (PAN 53) and late maturing (PAN 7M-83) varieties. Overall, the results showed that S. frugiperda induced slight leaf damage but detectable impact on yield at a specific plant developmental stage. This model will contribute to the development of decision-support tools for IPM. Overall, the topic is interesting and the data is well enough to support the main conclusion. However, the manuscript needs careful proofreading and revision. Grammar mistakes and poor writing are undermining the significance of this study. Besides, there are some major points which need clarifications before possible publication in PONE.

We thank the reviewer for their assessment of the MS and have made an effort to improve the readability of the MS. The points raised have been addressed below.

Introduction section

In line 59, I suggest providing more examples of studies describing host range and invasiveness of Spodoptera frugiperda, for example recent studies by Sokame et al. (2020), Wu et al. (2021) and Wang et al. (2022) can be cited.

* Sokame BM, Subramanian S, Kilalo DC et al. (2020) Larval dispersal of the invasive fall armyworm, Spodoptera frugiperda, the exotic stemborer Chilo partellus, and indigenous maize stemborers in Africa. Entomol Exp Appl 168:322–331.

* Wu PX, Ren QL, Wang W, Ma Z, Zhang RZ (2021a) A bet-hedging strategy rather than just a classic fast life-history strategy exhibited by invasive fall armyworm. Entomol Gen 4:337–344.

* Wang, P.; He, P.C.; Hu, L.; Chi, X.L.; Keller, M.A.; Chu, D. (2022) Host selection and adaptation of the invasive pest Spodoptera frugiperda to indica and japonica rice cultivars. Entomol. Gen. 42, 403–411.

We have added these references to the text.

In introduction section, a reference is missing to support the statement made is line 77, please revise as …provided by natural enemies (Croft 1990; Desneux et al. 2007).

- Croft, B. A. (1990). Arthropod biological control agents and pesticides. John Wiley and Sons Inc..

- Desneux, N., et al., 2007. The sublethal effects of pesticides on beneficial arthropods. Annu. Rev. Entomol. 52, 81–106.

We have added these references.

- L118: The authors should provide a schematic for the methodology section, which will help readers to easily understand the overall sketch of this study.

We have added a schematic to describe the experiment (Figure 1)

- “We inoculated plants with a single 2nd instar S. frugiperda larvae, which was placed in the maize funnel, at V5, V8, V12, VT and R1 maize growth stages” Why second instar? Please discuss it.

We now provide a explanation, as follows: “We elected to use 2nd instar larvae because (i) neonates disperse via ballooning and we wanted to be sure the innoculated plant received the treatment, and (ii) later instar larvae are more complicated to rear because FAW become cannibalistic from the 3rd instar.”

- “In addition, the height of the plant and the chlorophyll content were recorded.” How? Please add further details about this statement.

We have removed these details, as these data were not used in the final analysis.

- The data analysis section too much wordy and complicated. The authors should rewrite more clearly and concisely.

We have removed some statements from the Data analysis section to improve readability.

- In Discussion Section: “Not only is this potentially wasting limited resources, but promoting the use of toxic chemicals to poor farmers, who rarely use protective clothing nor understand how to safely use poisons, carries substantial risks to human health and the environment.” This sentence is very important. The authors should also add that the overuse/misapplication of chemicals insecticides have side effects on non-target species (Desneux et al. 2007; Palma-Onetto et al. 2021). Please cite these key articles.

- Desneux, N., et al., 2007. The sublethal effects of pesticides on beneficial arthropods. Annu. Rev. Entomol. 52, 81–106.

- Palma-Onetto, V., Oliva, D., González-Teuber, M. (2021). Lethal and oxidative stress side effects of organic and synthetic pesticides on the insect scale predator Rhyzobius lophanthae. Entomologia Generalis, 41 (4), 345-355.

We have added an explicit statement and these references.

- From sentences “To improve understanding of the relationship between leaf damage and yield loss, we performed a screenhouse experiment on three commonly used varieties in Zambia, including an early, medium and late maturing variety.” To “Typically pest resistance carries a yield penalty [44,45] and hence the use of S. frugiperda resistant or tolerant varieties may only be justified if the anticipated pest damage is very high, or if these attributes can be linked with others that enhance performance under small holder management” Both these paragraphs are redundant. The results are already presented in results sections. In discussion, the authors should compare and justify their main findings to recently published work.

We have shortened the text of these paragraphs and generally aimed at concise language. However, we do not regard these as redundant. They interpret the significance of our results within the broader context of smallholder FAW management. It was a surprise to us that such a small (3%) proportion of the variance in yield could be explained by FAW induced damage, despite the controlled screenhouse conditions. It is important that the implications of this for FAW management are discussed.

- The current version of discussion section is not okay. The authors should rewrite whole discussion section. Remove the redundant sentences, compare and justify your results with recently published related work from reputable journals.

As stated above, we have shortened the text and aimed at more concise language. We also added citations to other research, although despite the long history of FAW research there are relatively few studies that actually measure FAW impact on yields.

- The authors should merge the A, B, C, and D separated figures as Figure 2.

These should been presented as separate panels of the same figure. It was only our misunderstanding of the file naming in PONE that led to their presentation as separate figures.

References:

The ref 1 is outdated and should be deleted because the new reference by Kenis et al. (2023) is highly comprehensive and detailed.

We have remove Ref 1

L315, citation for reference 2 should be corrected as :

Kenis M, Benelli G, Biondi A, Calatayud PA, Day R, et al. (2023) Invasiveness, biology, ecology, and management of the fall armyworm, Spodoptera frugiperda. Entomologia Generalis doi.org/10.1127/entomologia/2022/1659

This citation was as generated using the PONE style. It is updated as the paper is now fully published.

6. PLOS authors have the option to publish the peer review history of their article (what does this mean?). If published, this will include your full peer review and any attached files.

Do you want your identity to be public for this peer review? For information about this choice, including consent withdrawal, please see our Privacy Policy.

Reviewer #1: No

Reviewer #2: No

---

## [Editor Report · Decision Letter 1]

25 May 2023

Understanding the impact of fall armyworm (Spodoptera frugiperda J. E. Smith) leaf damage

on maize yields

PONE-D-22-30949R1

Dear Dr. Harrison,

We’re pleased to inform you that your manuscript has been judged scientifically suitable for publication and will be formally accepted for publication once it meets all outstanding technical requirements.

Kind regards,

Nicolas Desneux

Academic Editor

PLOS ONE

---

## [Editor Report · Acceptance letter]

2 Jun 2023

PONE-D-22-30949R1 

Understanding the impact of fall armyworm (*Spodoptera frugiperda* J. E. Smith) leaf damage on maize yields 

Dear Dr. Harrison:

I'm pleased to inform you that your manuscript has been deemed suitable for publication in PLOS ONE. Congratulations! Your manuscript is now with our production department. 

Kind regards, 

on behalf of

Dr. Nicolas Desneux 

Academic Editor

PLOS ONE